# Hierarchical Multipath Blockchain Based IoT Information Management Techniques for Efficient Distributed Processing of Intelligent IoT Information

**DOI:** 10.3390/s21062049

**Published:** 2021-03-14

**Authors:** Yoon-Su Jeong, Sung-Ho Sim

**Affiliations:** 1Department of Information Communication Engineering, Mokwon University, Daejeon-si 35349, Korea; bukmunro@gmail.com; 2College of General Education, Semyung University, Jechon-si 27136, Korea

**Keywords:** distributed cloud, integrity, IoT, blockchain, probability, security

## Abstract

As cloud technology advances, devices such as IoT (Internet of Things) are being utilized in various areas ranging from transportation, manufacturing, energy, automation, space, defense, and healthcare. As the number of IoT devices increases, the safety of IoT information, which is vulnerable to cyber attacks, is emerging as an important area of interest in distributed cloud environments. However, integrity techniques are not guaranteed to easily identify the integrity threats and attacks on IoT information operating in the distributed cloud associated with IoT systems and CPS (Cyber-Physical System). In this paper, we propose a blockchain-based integrity verification technique in which large amounts of IoT information processed in distributed cloud environments can be guaranteed integrity in security threats related to IoT systems and CPS. The proposed technique aims to ensure the integrity of IoT information by linking information from IoT devices belonging to subgroups in distributed cloud environments to information from specific non-adjacent IoT devices and blockchain. This is because existing techniques rely on third-party organizations that the data owner can trust to verify the integrity of the data. The proposed technique identifies IoT information by connecting the paths of IoT pre- and subsequent blocks into block chains so that synchronization can be achieved between subgroups in distributed cloud environments. Furthermore, the proposed technique uses probabilistic similarity information between IoT information blocks to react flexibly to subgroups that constitute distributed clouds so that IoT information blocks are not exploited maliciously by third parties. As a result of performance evaluation, the proposed technique averaged 12.3% improvement in integrity processing time over existing techniques depending on blockchain size. Furthermore, the proposed technique has to hash the IoT information that constitutes a subgroup with probability-linked information, validating the integrity of large-capacity IoT information, resulting in an average of 8.8% lower overhead than existing techniques. In addition, the proposed technique has an average improvement of 14.3% in blockchain-based integrity verification accuracy over existing techniques, depending on the hash chain length.

## 1. Introduction

With the growing demand for services and the emergence of double-ear various forms of services, distributed cloud environments have changed from centralized to decentralized as a way to meet requirements such as real-time and low-latency requirements [1]. The distributed cloud environment is easy to analyze and utilize big data and features the ability to freely expand the infrastructure needed for analysis. Additionally, distributed clouds are divided into public and private clouds depending on where the data is stored, and blockchain is often applied to distributed cloud environments for the safe processing of big data. Blockchain is applied because blockchain has good security and can reduce the cost of protecting data [2,3].

In a distributed cloud environment, the use of cloud services is steadily increasing, focusing on portable IoT (Internet of Things) devices (such as mobile phones, tablets, and small devices) rather than personal PCs or servers, and the requirements for integrity verification process for information sent and received from IoT devices are increasing. This is because most of the IoT information used in distributed cloud environments has so far no clear security measures in place to ensure the integrity and safety of IoT information. IT companies (MS, Amazon, IBM, etc.) related to IoT devices are preparing blockchain-based cloud services in recognition of the importance of the integrity of information sent and received from IoT devices [4].

Recent research has leveraged new ideas in cloud services that utilize the blockchain to distribute and track blockchain to efficiently audit data processed in a cloud environment [5]. However, existing studies do not have a certain probability of fake names being tracked by names, so blockchain alone can only operate cloud data, resulting in increased computational overhead and economic cost losses from using smart contracts.

In this paper, we propose a blockchain-based IoT big data integrity verification technique that can ensure the integrity of IoT information so that large amounts of information sent and received from heterogeneous IoT devices are safely sent to cloud servers (data centers). The proposed technique ensures the integrity of IoT information by connecting information from specific non-adjacent IoT devices (emergency data that needs to be immediately undertaken and data sent to the data center for subsequent measurement and use) to blockchain-based multiple hash chains. This is because existing techniques rely on third-party organizations that the data owner can trust to verify the integrity of the data. Furthermore, the proposed technique prevents the malicious use of IoT information blocks to third parties by linking the paths of IoT information transfer blocks and subsequent blocks to blockchains so that only the exact desired information can be determined from a large amount of IoT information. The proposed technique uses probabilistic similarities of IoT information blocks so that they can respond flexibly to changes in the subgroups that make up the distributed cloud. Furthermore, the proposed technique determines whether to add IoT block information to the blockchain connected to the lower group, verifying the hash values stored in the blockchain, thus ensuring synchronization as well as reliability.

The proposed technique has the following four objectives to ensure the integrity of blockchain-based IoT big data optimized for distributed cloud environments:

First, the proposed technique uses blockchain and multiple hash chains for IoT information linkage to maintain synchronization between subgroups in distributed cloud environments.

Second, the proposed technique dynamically operates the IoT information processing process according to contextual information in the distributed cloud environment using hierarchical multi-step methods to retrieve IoT information accurately and quickly.

Third, the proposed technique improves the possible load and throughput rates between IoT devices by requiring the distributed deployment of IoT information blocks of lower groups in distributed cloud environments to be uniformly distributed at n-bit sizes.

Fourth, the proposed technique improves the throughput rate of IoT information by transferring all IoT information with a difference in the IoT information transfer interval, instead of continuously transferring all IoT information to minimize processing time.

The remainder of this paper is organized as follows. Section 2 explores blockchain and existing research. In Section 3, an efficient integrity verification technique using blockchain-based hierarchical IoT keys is proposed, and Section 4 performs a performance evaluation of the proposed and existing techniques. Finally, we conclude in Section 5.

## 2. Preliminaries

### 2.1. Blockchain

Blockchain technology is a technology that provides high security without the need for centralized institutions with reliability as collateral. As in Table 1, the distributed ledger used in the blockchain is a distributed digital ledger that discloses content to all transaction members (participants), unlike the traditional method of keeping transaction details in a central ledger when a transaction occurs. The centralized ledger management technology is managed by a third party organization, with the central management of the ledger. That is, distance details are managed through a third party that has been proven to be a reliable institution.

Blockchain has advantages such as reduced service development time and ease of development when used in distributed cloud environments. Table 2 illustrates the characteristics of using blockchain in distributed cloud environments.

Blockchain technology has become an opportunity to change the perception of value and utilization as e-money due to the success of Bitcoin. However, blockchain technology provides high security in the field of e-financial transactions as well as electronic currencies, resulting in significant savings in operating costs. Various benefits can be gained, especially if blockchain services are constructed based on existing cloud environments.

### 2.2. Related Works

#### 2.2.1. Traditional Work on Data Integrity

In a distributed cloud environment, heterogeneous devices cannot be completely controlled because different types of information are sent and received to receive cloud services. Several warranty models have been proposed in recent studies to address these challenges for cloud services [2].

Bowers et al. have proposed that the retention of cloud customer data can always be verified by the record holder and facilitates certifiers easily as open key isomorphisms [6]. This technique exchanges conformity-approval ventures with external reviewers so that they cannot pick up data information from cloud customers to save on processing assets from neighboring customers.

Erway et al. established a PDP (Provable Data Possession) model that guarantees the integrity of the data in storage outsourcing related to model setup verification while third parties act as validators [7]. This model not only allows cloud service providers to jointly store and maintain data compared to existing models but also allows information owners to easily change archives.

Wang et al. proposed a model in the publication cloud that performs retention checks on behalf of data held by remote data customers [8]. Based on the bilinear pairing technique, the model designed its protocol to efficiently perform surrogate data retention testing and efficiently demonstrated safety and security analysis.

Zhu et al. proposed an efficient way to select parameter values to optimize homomorphic verification responses and hash index hierarchies to ensure data integrity in storage outsourcing [9]. The proposed method provides low open and communications overhead so that multiple cloud service providers can jointly store and maintain customer data.

Tenies et al. proposed the concept of an available data subsystem (PDP) that can safely verify the integrity of data at remote locations in a cloud environment [2]. However, we fail to demonstrate that the integrity of the data is safe in real-world cloud environments.

Wang et al. proposed a model that applies the concept of TPA (Third-Party Agent) to the PDP scheme [10]. However, this model can reduce the user burden by allowing TPA to perform audits on behalf of users, but data leakage problems can arise because TPA cannot be trusted in the PDP scheme.

#### 2.2.2. Application on Data Integrity

Nguyen et al. proposed a confidence evaluation model to reconsider the value of confidence over some time in the process of investigating the device’s behavior to create knowledge of the device [11]. The model is characterized by measurements based on the probability that the level of uncertainty in each challenge–response interaction will respond as expected by the device or with an expected response.

Gwak et al. proposed a new authentication technique for users accessing public systems based on the psychological concept of eye-care [12]. This technique uses a mechanism to share the rights granted to users during the initial authentication phase of the user’s authentication process. Additionally, psychological concepts were used to identify users whose roles and similarities coincide [13].

Amazon et al. proposed an authentication protocol using physical non-clone functions (PUFs) to improve the power and processing power of IoT systems [14]. The protocol features a challenge–response approach between IoT devices, facilitating safe session setup.

Dukas et al. proposed PKI (Public Key Infrastructure) techniques for IoT devices for securing PKI-based wearable IoT data integrity [15]. This technique guarantees the safety of data using digital signatures or encryption techniques for private and public key encryption based on PKI.

To reduce the loss of data in high-frequency applications, Aman et al. proposed a physically non-clone random-time hop-based technique [16]. This eliminated the need to store the keys locally on the device to ensure physical safety. In addition, it is characterized by verifying the integrity of multiple packets by combining packets with random time shifts and random permutations to ensure safety against physical attacks.

Conti et al. proposed a technique that optionally filtered the data processing and encryption capabilities of the edge nodes to decentralize the sense generation burden in a centralized approach [17]. This technique disperses the burden of data generation by reducing or limiting data volumes to efficiently process in-device data analysis. However, the problem is that this technique has no mechanism to protect data at the edge nodes, making it vulnerable to security.

#### 2.2.3. Blockchain-Based Work on Data Integrity

Wang et al. proposed an isomorphic authentication ring signing technique to protect the privacy of the users who make up the cloud environment [18]. However, this technique has a problem that is not appropriate in a cloud environment where large-scale users demand services because of the cost.

Huang et al. proposed a model using multiple TPAs to solve the problem of collusion and centralization in a cloud environment [19]. However, the model is used by adding centralized receiving servers that can learn the identity of users.

Liu et al. proposed a model that addresses the problem of malicious TPA trying to steal data under a false identity in a cloud environment [20]. However, this model has the problem of not being able to audit the identity of malicious TPA if the number of components that make up the cloud is large.

Liu et al. proposed an IoT data integrity technique based on the blockchain of cloud storage service architecture and data integrity services [21]. This technique dynamically configures the IoT environment to perform integrity without relying on TPA. However, the IoT data upload speed and data size have the challenge of improving to suit the cloud environment.

Liang et al. proposed a data validation technique that distributed the cloud environment to validate the data of the devices that make up the cloud environment [22].

Yue et al. proposed a framework using blockchain to ensure the data integrity of P2P cloud storage [23]. This technique is characterized not only by analyzing the performance of the system using the Merkle tree structure but also by being optimized for the P2P environment to ensure the integrity of the data.

Wang et al. proposed a model that decentralized the cloud environment to change the single point of trust problem, a problem with existing data audit service models, to group trust [24]. The model ensures the integrity of the data through collective trust, allowing users to track their data in the model.

#### 2.2.4. Others Work on Data Integrity

Izoninetal. proposed a novel ensemble of neural network tools that improves the accuracy of predictive tasks for recovering missing IoT data [25]. The proposed technique is characterized by two consecutive general regression neural network networks and a continuous geometric transformation model being constructed in one neural similar structure, replacing the sum of the results of the general regression neural network with a weighted sum, improving the accuracy of the operation. However, the technique failed to improve the operational efficiency of IoT-based systems because it fails to consider the possibility of designing AIoT (Artificial Intelligence of Things)-based hardware variants.

Zare et al. proposed a method to determine tunneling resistance in some samples and to represent tunneling resistance of polymer nanocomposites to estimate the inspiration of different parameters in tunneling resistance [26]. This method uses low tunneling distances and small penetration thresholds to reduce tunneling resistance and obtain conductive nanocomposites. However, it has been shown that the concentration and length of carbon nanotubes affect tunneling distance and interface tension among the parameters used in the study.

Tkachenko et al. proposed a method that can effectively handle missing data collected by a specific sensor by using a general regression neural network [27]. The method uses two sequentially connected general regression neural networks to ensure optimal parameters are selected. However, the method does not analytically demonstrate the approximation and partial elimination potential of computational intelligence errors in large numbers of different IoT collection data, although the high accuracy of cascade operations is inserted compared to existing methods.

Dong et al. [28] conducted a protocol-related study for data integrity check in a cloud environment. Dong et al. analyzed the requirements and problems of state-of-the-art protocols for integrity checks of data in cloud environments. Furthermore, we compare the advantages and disadvantages of protocols used in cloud environments with each other.

Pujar et al. studied data integrity and validation, including data integrity checks, performance metrics, security attacks, and updates for cloud storage [29]. Specifically, the taxonomy developed based on security-level provisioning used in various techniques for data integrity and validation techniques provides a global view of the problem and solution.

Maheswari et al. extended the cloud platform to study various integrity verification methods for IoT applications to be applied to both cloud and fog environments [30]. In particular, Maheswari et al. described the authentication structure and its limitations being used in existing integrity verification protocols.

## 3. Distributed Management of Hierarchical IoT Information Based on Blockchain

The cloud environment is making various efforts to meet the requirements of real-time and low-latency IoT devices as various types of application services are developed along with the growing demand for IoT devices held by users. However, as the cloud environment becomes huge, distributed cloud environments have been built and utilized as a way to maximize the management efficiency of IoT devices, but there is no clear way to safely verify the integrity of the information that IoT devices send and receive in distributed cloud environments. Recently, blockchain-based cloud services are being prepared to ensure the integrity of IoT information, focusing on large enterprises, but they do not guarantee the integrity of all IoT devices of this species. In this section, we propose a blockchain-based IoT integrity verification technique to ensure the integrity of large IoT devices in a distributed environment. The proposed technique performs hierarchical grouping of IoT information into multiple hash chains so that IoT information in layer n can be cross-validated in two directions with IoT information in layers *n*−1 and *n* + 1 to prevent the problem of IoT information from being corrupted or deleted. The proposed technique ensures the integrity of IoT information by verifying the hash values stored in the blockchain as cumulative probability values.

### 3.1. Overview

To efficiently manage information transmitted and received from heterogeneous IoT devices, the cloud environment manages IoT devices by separating the cloud in a distributed manner, so safety problems are more serious than wired environments. Additionally, the distributed cloud environment is not guaranteed to ensure the integrity of IoT information as the latest technologies such as IoT devices and big data are being used indiscriminately. The proposed technique uses cumulative probability values generated by applying arbitrary random keys and weight information to blockchain to cross-validate heterogeneous IoT information operating in distributed cloud environments in two directions, thereby reducing the distributed IoT information processing load as well as ensuring the integrity of IoT information.

To efficiently verify the integrity of IoT information, the proposed techniques have the following purposes.

First, the proposed technique validates the integrity of IoT information in a hierarchical distributed network structure that does not rely on the authority of the central server to minimize the bottleneck of IoT devices.

Second, the proposed technique performs two-way verification by constructing arbitrary random keys and signature keys generated at layers *n* − 1 and *n* + 1 in the form of blockchain to verify the integrity of IoT information.

Third, we use multiple hash chains to apply weight information on IoT information to blockchain technology. The reason for using blockchain-based multi-hash chains in the proposed technique is to safely verify IoT information distributed in a hierarchical structure.

The proposed technique group IoT devices in a hierarchical structure as shown in Figure 1, then generates signature keys for each group grouped. As shown in Figure 1, the proposed technique selects an intermediate party that collects and delivers important information from IoT devices composed of hierarchical structures. Intermediate parties can also pass collection information directly to the server, but the intermediate parties in the adjacent group are passed to the server in a unified hop-by-hop. The proposed technique applies weight information of IoT devices to blockchain technology to ensure the integrity of IoT devices by applying cumulative probability values to weights. By maintaining the weights of IoT devices allocated with cumulative probability values at regular intervals, we can maintain and verify synchronization between the server and IoT devices, IoT devices, and IoT devices.

To efficiently group IoT information, the proposed technique simplifies the importance of IoT information into a multi-layer structure like Figure 2 so that it can be computed hierarchically using blockchain, and then applies the importance of IoT information to the pairwise comparison matrix. The pairwise matrix information is represented to enable the number of comparisons as many times as n(n−1)/2 times of the total information of IoT information. The reason is to group IoT information hierarchically to facilitate IoT information management.

In Figure 2, the importance of IoT information is multi-processed with all information associated with IoT information using blockchain-based hash functions ($H_{IoT}$: {0,1} →
$Z_{N}$), as shown in Equation (1):(1)$$H_{IoT}:\;{\left\{0,1\right\}}^{*}\times Z_{N}\to Z_{N}.$$

The proposed technique pairs IoT information with devices that utilize IoT information using expression (2) so that it is located in the k-th layer. In Equation (2), IoTI refers to the $n_{i-1}$
·
$n_{i}$ matrix that constitutes the $\omega_{i}$ vector among the information estimated from IoT information. $n_{i}$ means the number of IoT information in the i-th layer:(2)$$\mathrm{IoTI}\;\left[1,\;k\right]\;=\;\prod_{i=2}^{k}\mathrm{IoTI},$$
where k stands for a layer of information collected from IoT devices.

In the proposed technique, IoT information is tied hierarchically according to the weight information of IoT information, such as Figure 3, so that it can be handled accurately in a short time by combining blockchain and non-blockchain. This process is because large-scale IoT information can be quickly verified in a short time. The IoT information in Figure 2 binds IoT attribute information $v_{t}$ with hash chains to select seed ${\hat{y}}_{t}$ according to similar information $h_{t}$ in IoT.

In Figure 3, IoT information paths and core extraction processes group IoT information in a ranking with high IoT information association (Ranking). At this time, the proposed technique examines the similarity with the connectivity group of IoT information, and groups the possible entire path into one group. This is because patterns with the relatively high association can predict different paths as patterns of IoT information in the future. The proposed technique applies the weighted probability of hierarchically constructed IoT information to hash chains without additional information based on the theory of weight probability, such as expression (3):(3)$$P_{xy}\left({\mathrm{IoT}}_{i}^{t-1},\;{\mathrm{IoT}}_{i}^{t}\right)\;=\;P_{x}\left({\mathrm{IoT}}_{i}^{t-1}\right)\;P_{y}\left({\mathrm{IoT}}_{i}^{t}\right)\;i,j=1,\cdots ,n.$$

The proposed technique uses seed values to perform synchronization to apply weights to a large number of IoT information and then processes IoT information at layer *t* between layer n+1 and layer n−1. This is because we apply arbitrary random key and weight information to the blockchain to ensure the integrity of the IoT information by using cumulative probability values so that it can be cross-validated in both directions when layer-wise separated IoT information is linked together.

### 3.2. Link IoT Devices between Subgroups

The proposed technique uses a multi-path hash chain as shown in Figure 4 to authenticate the in-group IoT information to synchronize IoT devices between subgroups. As IoT devices are added to the distributed cloud environment, we require authentication to separate the authentication path to minimize overhead into two (primary and secondary paths). The purpose of the main path during the authentication path is to authenticate all connected IoT devices, and the purpose of the auxiliary path during the authentication path is to check the authentication and integrity between IoT devices. The proposed technique validates the integrity of IoT information through information from auxiliary paths.

To merge IoT devices in a group, we divide them into even and odd numbers to efficiently check their integrity according to the number of IoT devices, such as expressions (4) to generate hash values of the auxiliary paths associated with the previous IoT devices:(4)$$k_{n}=\begin{array}{l} h(k_{n}\|x_{n-1})\;n=\mathrm{ood}\\ h(k_{n-1}\|y_{n})\;n=\mathrm{even} \end{array},$$
where $x_{n-1}$ and $y_{n}$ mean before and after IoT information in the group, and the hash value $k_{n}$ is generated according to the number of IoT information n.

To check the integrity between IoT information at expression (4), the auxiliary path $x_{n}$ associated with the previous IoT information is used for IoT information cross-information authentication by applying it to the hash function along with the previous hash value $k_{n+1}$. Additionally, the hash value $k_{n}$ for the main path of IoT information is used with auxiliary paths $x_{n}$ and $y_{n}$.

### 3.3. Creating IoT Subgroup Keys Using Vector Approximation

The proposed technique can vector random-collar information, which is probabilistically extracted from the information of IoT devices that make up the distributed cloud environment, to generate keys that efficiently verify the integrity of IoT information by organizing it as a sum of orthogonal vectors.

Figure 5 shows the process of generating signature keys for subgroups to verify the integrity of IoT information in the proposed technique. In Figure 5, the proposed technique obtains vector blocks $cb_{i}$ by selecting m of any of the n different orthogonal color information vector blocks $x_{1}$, $\;x_{2}$, ⋯, $x_{n}$ as Expression (5):(5)$$cb_{i}≅\sum_{i=1}^{m}(\sum_{j=1}^{n}(c_{i}x_{j})),$$
where vector block $cb_{i}$ uses the only $c_{1}$, $\;c_{2}$, ⋯, $c_{m}$ values to become an identity, no longer approximation.

The error in approximating the vector block $cb_{i}$ as a result of the Expression (5) is as shown in Expression (6):(6)$$e\;=\;cb\;-\;\sum_{i=1}^{m}(\sum_{j=1}^{n}(c_{i}x_{j})),$$
where error e means a value perpendicular to the plane $x_{1}$, $x_{2}$, ⋯, $x_{m}$, and $c_{1}x_{1}$, $c_{2}x_{2}$, ⋯, $c_{m}x_{m}$ means a value where error e is minimal when calculating vector block *cb* over $x_{1}$, $x_{2}$, ⋯, $x_{m}$.

In Figure 3, the key attribute information of IoT information, which consists of hierarchical subgroups, consists of hierarchical hash chains to obtain probability **p**, as shown in Expression (7), and then link the random-color information, which is probabilistically extracted from the information of IoT devices, with IoT attribute information:(7)$$p\;=\{\begin{array}{ccc} \frac{n}{L}-\frac{n{\left(\left(L-1\right)!\right)}^{2}}{L^{2}\left(L-21\right)!\left(L-1\right)!}\times \frac{1}{100} & & L\geq 21\\ \frac{n}{L}\times \frac{1}{100} & & 1\leq L<21 \end{array}\;.$$

The reason why we use the connection probability **p** of IoT attribute information, such as expression (7), for probability-extracted random-collar information is to enable accurate extraction and verification of IoT attribute information with high probability values.

To generate the key for the best n+1 layer in Figure 5, we generate signature keys such as expression (8) using random vector block cb of IoT color information vector blocks $x_{1}$, $\;x_{2}$, ⋯, $\;x_{n}$ and selected seed $r_{1}$ from ${}_{R}{\left\{0,1\right\}}^{n}$ and connection probability **p** of IoT attribute information:(8)$$MKey\;=\;{\left(-1\right)}^{r_{1}}\cdot h^{n}(cb,p)\;\mathrm{mod}\;N.$$

The *MKey* generated by expression (8) is encrypted and shared as shown in expression (9) using the shared key SK between subgroups to maintain synchronization with other subgroups:(9)$$\mathrm{Transfer}E_{SK}(cb_{i},\;MKey,\;p,\;r_{1}).$$

The proposed technique, via *Mkey*, cannot only simplify the classification and management of complex information transmitted and received from IoT devices by binding IoT information links into a bullock chain but also confirms the association and similarity of IoT information.

### 3.4. Creating and Validating IoT Information Blocks

The proposed technique is performing with algorithms 1 and 2 to generate and validate blocks of IoT information processed in authentication paths (primary and secondary paths) in the form of blocks with multi-path hash chains.

In Table 1, the IoT information block generation algorithm randomly selects IoT information from the lower group after storing IoT information and generates a new block from ${}_{R}{\left\{0,1\right\}}^{n}$. When the new block is the first block, the previous block hash is set to zero. When the first block is generated, the hash value is generated by dividing the path into even and odd numbers to efficiently check its integrity according to the number of IoT information in the lower group. If it fails to check the integrity of IoT information, it is required to reconstruct the blocks of IoT information.
**Algorithms 1** Block generation algorithm for IoT information.**Input: IoT information included in the subgroup** **Output: Generate replication information for odd/even** 1: **for** all IoT information do 2:     The IoT device generates IoT Information *i*
∈ [1, *k*] 3:      Check its IoT Information 4:      **if** sub-group generate IoT Information from IoT devices **then** 5:         Store the IoT Information block 6:       **else** 7:          **Return**
*i* 8:       **end if** 9:       **if**   the IoT Information block can be identified **then** 10:         Regenerate hash values for odd/even of IoT Information 11:      **else** 12:          reconfirm block the IoT Information 13:      **end if** 14: **end for**

In Algorithm 2, IoT information checking algorithms can reduce the overhead of IoT devices by adding two hash values generated by the number of replications of odd/even to the first and last of multiple hash chains to verify the integrity of IoT information.
**Algorithm 2** Check algorithm of IoT information block.**Input: IoT information block** **Output: Results of IoT information block check in the subgroup** 1: share its block between each subgroup 2: **for** all IoT block information do contain from IoT device 3:     **if** contain its block information in the subgroup **then** 4:         **if** all the IoT block information are identical **then** 5:             Accept this block 6:             broadcast the accepted block information 7:          **else** 8:              Reject this block 9:              Reject broadcast the rejected block information 10:           **end if** 11:     **end if** 12:  **end for** 13:  **if**  accepted block information > rejected block information **then** 14:          Each IoT device stores this block 15:  **else** 16:          Each IoT devices delete this block 17: **end if**

By Algorithm 2, the proposed technique groups IoT information to any size and stores it as a blockchain with the access control policy of IoT information, so it compares the blockchain with the non-blockchain of IoT information and updates the blockchain. To save storage space for each IoT block information, each IoT block information can be periodically erased after uploading it to a cloud server.

The algorithm 2 can perform error judgments on large-capacity IoT information blocks through connection errors in IoT block information, which can improve connection accuracy for IoT block information.

## 4. Evaluation

### 4.1. Environment Setting

The environment setting for the performance evaluation of proposed techniques is as shown in Table 3. Experiments of the proposed technique use two modes: Mode 0 and Mode 1. Mode 0 means that all transcoders are offloaded to the server with work being provided. Mode 1 means that all transcode is offloaded to neighboring IoT groups. The network range of performance evaluation is set to 400 m considering the IoT transmission and received range, and the available bandwidth of Server and IoT is set to 12 MHz/6 MHz. The maximum size of the blockchain is set to 2 Mbytes. In simulations, each IoT device that constructs a distributed network generates packets arbitrarily and transmits them to other IoT nodes. It is assumed that all IoT information packets are transferred to the relative IoT device in time and that all IoT devices can be synchronized. Other simulation parameters are listed in Table 4.

The proposed technique uses Arduino devices such as Figure 6 and Raspberry devices for IoT equipment built-in experimental environments. Figure 6’s IoT equipment, built-in experimental environments, is designed to be transmitted and received via Bluetooth or Wi-Fi. The programming language used for implementation in the algorithm of the proposed technique was C/C++.

### 4.2. Performance Analysis

#### 4.2.1. Blockchain Creation Time According to Blockchain Length

Table 4 is a result of comparing blockchain generation time over blockchain length using the probability of IoT information (date, time, size, purpose, etc.) when creating a blockchain-based hierarchical cloud group. Experiments in Table 4 show that the proposed technique for linking user information with each other as a probability value of IoT information distributed in the overlay network in the cloud environment is an average 7.9% reduction from the case where authentication information is not generated by the transaction. This result is because each group-specific Region Certification Authority (RCA) node generates blocks only when the transaction is of a constant size, so that the large number of IoT information processed in the cloud environment is easily controlled. This is because RCA is intended to improve accessibility to user information, and this process results in user information accurately in normal blocks over time.

#### 4.2.2. Latency Time Due to IoT Information Processing in a Blockchain-Based Overlay Network

Table 5 is a result of comparing latency according to IoT information verification processing of blockchain-based overlay networks. Experiments on Table 5 show that the proposed technique averaged 13.3% lower latency compared to methods without overlay networks. These results are the result of the proposed technique not only linking the blockchain of stochastic weighted IoT information to different probability elements, but also improving the accessibility of IoT information. To minimize the network load, the proposed technique shares IoT information by blockchain and excludes IoT information that is not authenticated by RCA within the group to blockchain generation.

#### 4.2.3. Comparison of Integrity Verification Processing Times by Blockchain Size

The processing time required for integrity verification according to blockchain size obtained the same result as Table 6. Table 6 validates the integrity between IoT devices according to their blockchain size when a subgroup of distributed clouds is configured by IoT devices. In particular, IoT devices that makeup subgroups build subgroups based on hierarchical properties and probability-linked information values, so the size of subgroups depends on the number of information of IoT devices included in the subgroups. Experimental results from Table 6 show that the proposed technique achieves an average 12.3% improvement in integrity processing time over existing techniques depending on blockchain size. These results are because the proposed techniques used the hierarchical properties of IoT information and the probability-linked information values to generate subgroups when generating them.

#### 4.2.4. Blockchain-Based Integrity Verification Overhead Comparison

Table 7 shows the integrity verification overhead of IoT devices that occur in processing IoT information connected by blockchain according to the size of the subgroups that make up the distributed cloud environment. As a result of Table 7, the proposed technique has obtained results with an average of 8.8% lower overhead than conventional techniques because it hashes the IoT information that constitutes subgroups with probability-linked information to verify the integrity of high-capacity IoT information. These results are based on the fact that the number of IoT devices in a distributed cloud environment is divided by an even number and an odd number, and the hash value of the auxiliary path associated with the previous IoT device is linked with the probabilistic random color information extracted.

#### 4.2.5. Comparison of Blockchain-Based Integrity Verification Accuracy

Table 8 shows the accuracy of verifying the integrity of IoT information by linking IoT information to the blockchain when generating subgroups in distributed cluster environments. As a result of the performance evaluation of Table 8, the proposed technique resulted in an average 14.3% improvement in blockchain-based integrity verification accuracy over existing techniques depending on hash chain length. These results are based on the fact that the proposed technique leaked seed values of inter-subgroup synchronization via probabilistic linkage information of IoT information that constitutes subgroups.

#### 4.2.6. The Rate of Change in Key According to the Blockchain-Based Hash Value

Table 9 shows the average key change probability with changes in keys generated by the generators of hash chains, the standard variance, and the probability of mean key change when generating IoT information in a distributed cloud environment with hash chains. Expression (10) to Expression (13) was used in simulations by defining key lengths, key conversion probabilities, standard distribution and distribution probabilities of blockchain key numbers, etc. for blockchain environments:(10)$$\bar{BK}=\frac{1}{N}\sum_{i=1}^{N}BK_{i},$$(11)$$ABKP\left(\%\right)=\frac{\bar{BK}}{128}\times 100\%,$$(12)$$∆BK=\sqrt{\frac{1}{N-1}\sum_{i=1}^{N}{\left(BK_{i}-\bar{BK}\right)}^{2}},$$(13)$$∆ABKP=\sqrt{\frac{1}{N-1}\sum_{i=1}^{N}{\left(\frac{BK_{i}}{128}-P\right)}^{2}}.$$

As a result of Table 9, the average number of key lengths transformed by the hash chain key length of the IoT information contained in the lower group was shown closely at 128 bits over N iterations, with very low variance values equivalent to 128 bits.

## 5. Conclusions

As IoT technology advances, various environments related to cloud services are developing. There is an increasing number of cases in which third parties illegally exploit large amounts of information sent and received in small devices such as IoT devices. In cloud environments, technologies are required to distribute large amounts of IoT information to ensure the integrity of IoT information. In this paper, we propose a technique to ensure the integrity of IoT information that connects information from non-neighboring specific IoT devices within a subgroup based on blockchain. The proposed technique selects only the desired information from the IoT information so that it is not exposed to third parties, creating the path to IoT information transfer blocks and subsequent blocks. Furthermore, the proposed technique improves the reliability of IoT information by stochastically applying similarity of IoT information blocks to the blockchain of IoT information within subgroups. Furthermore, the proposed technique determines whether to add IoT block information to the blockchain connected to the lower group, validating the hash values stored in the blockchain. As a result of performance evaluation, the proposed technique averaged 12.3% improvement in integrity processing time over existing techniques depending on blockchain size. Furthermore, the proposed technique has averaged 8.8% lower overhead than existing techniques, as it has hashed the IoT information that constitutes a subgroup with probability-linked information to verify the integrity of large-capacity IoT information. The proposed technique improves the integrity verification accuracy of blockchain-based by an average of 14.3% over existing techniques, depending on the hash chain length. Furthermore, the proposed technique was able to confirm that the number of transformed average key lengths over hash chain lengths in the IoT information contained in the subgroup was close to 128 bits when the number of average key lengths was repeated N times, and the variance corresponding to 128 bits was very low. The proposed technique should consider the possibility of applying similarity of IoT information blocks to blockchain of IoT information within subgroups probabilistic when numerous different IoT information is collected. In future research, we plan to evaluate whether IoT information is exposed to third parties based on the results of this study by applying it to the actual operating cloud environment.

## Figures and Tables

**Figure 1 sensors-21-02049-f001:**
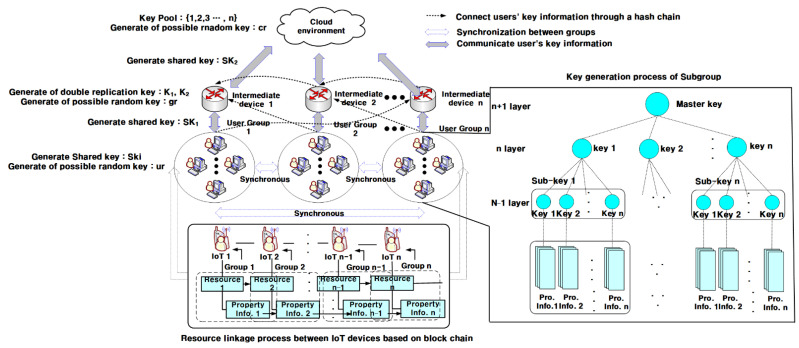
The overall behavioral structure of the proposed technique.

**Figure 2 sensors-21-02049-f002:**
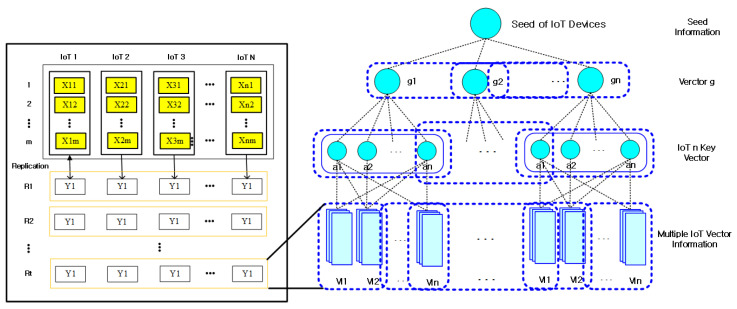
IoT (Internet of Things) information extraction using blockchain.

**Figure 3 sensors-21-02049-f003:**
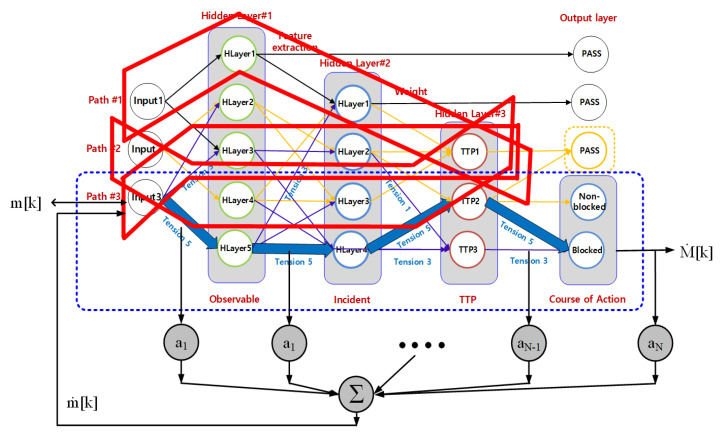
Layer processing with multiple hash chains.

**Figure 4 sensors-21-02049-f004:**
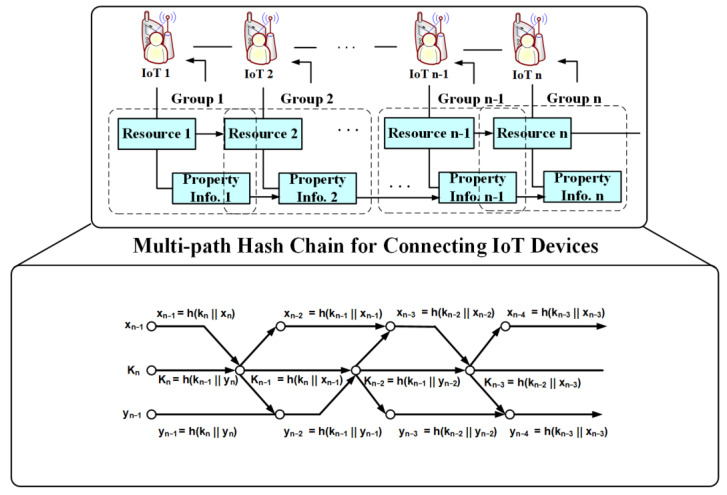
Multi-path hash chain for connecting IoT devices.

**Figure 5 sensors-21-02049-f005:**
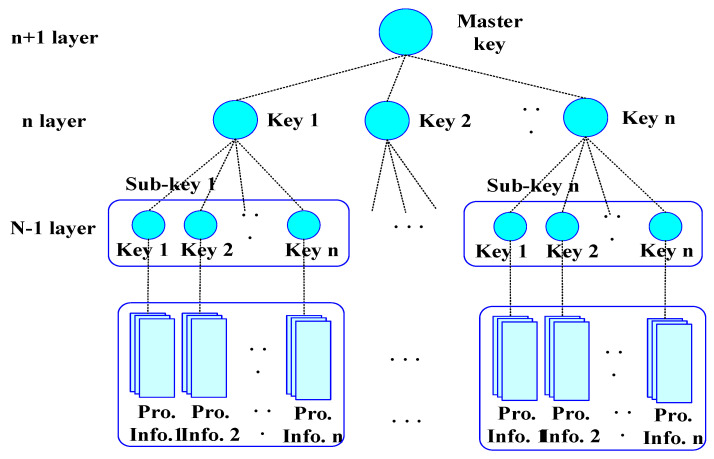
Process for creating subgroup keys of IoT.

**Figure 6 sensors-21-02049-f006:**
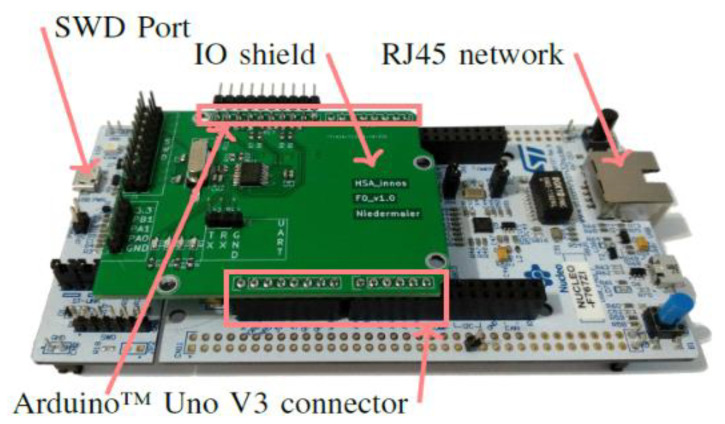
IoT device for simulation.

**Table 1 sensors-21-02049-t001:** Comparison of centralized ledger management technology with a distributed ledger management structure.

Division	Centralized Ledger Management	Distributed Ledger Management
Cloud Storage	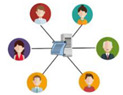	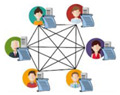
Type	Centralized management	Distributed management
Notarization and management entity	Notarize all transaction details by the central third party	-All transaction participants view, notarize, and manage transaction details -Transaction history is shared and archived for all network participants
Cost	The high cost of maintenance(management)	-Low system deployment costs -Low maintenance costs
Characteristics	-Advantages: (1) quick transaction speed, (2) ease of control -Disadvantages: Vulnerable to security (Dos vulnerable to attacks and hacking)	-Advantages: (1) Maintaining transparency in transaction information, (2) No DDos attack, (3) No forgery of transaction details. -Advantages: (1) relatively slow transaction speed, (2) the complexity of control

**Table 2 sensors-21-02049-t002:** Distributed cloud-based blockchain service features.

Feature	Description
Seamless blockchain provisioning	Provides a very simple model for creating blockchain as part of a PaaS environment.
Elastic scalability	Simplifies adding and removing nodes to a blockchain network.
Global Availability	Cloud environments enable blockchain provisioning anywhere in the world.
Simple programming model	Provides a simple programming model for creating blockchain applications by abstracting the underlying blockchain infrastructure.

**Table 3 sensors-21-02049-t003:** Environment setup.

Parameter	Value
The transmit/receive the power of the users $P_{U}$	0.1 W/0.05 W
The network coverage radius	400 m
The static circuit power $P_{C}$	0.1 W
The path loss exponent α	3
The available bandwidth for $\beta_{S}/\beta_{U}$	12 MHz/6 MHz
The maximum size of the blockchain	2 Mbytes
The power of noise $\sigma^{2}$	−174 dBm/Hz
The mean value of Rayleigh fading μ	1
Input data size $D_{m,\;n}$	5 kbits/s
Delay threshold $\tau_{m,n}$	10 s
Computation workload/intensity $X_{m,n}$	18,000 CPU cycles/bit
Computation energy efficiency coefficient of the processor’s chip in the APs/users $\tau_{m}^{A}$, $\tau_{m,n,k}^{U}$	$10^{-26}$
The computational capability of the Aps $f_{m}^{A}$	10–100 GHz CPU cycles/s
The Computational capability of the users $f_{m,n,k}^{U}$	1–10 GHz CPU cycles/s
The unit price of energy $\varphi_{e}$	0.1 Token/J

**Table 4 sensors-21-02049-t004:** Rate of change according to blockchain length.

Number of Blockchain	Blockchain Generation Time (ms)			
	No Using Transaction Cumulative	Using 3 Average Transaction Cumulative	Using 5 Average Transaction Cumulative	Using 10 Average Transaction Cumulative
10	5.71	4.82	3.53	2.45
25	6.43	5.27	4.39	3.26
50	7.12	6.68	5.23	4.48
100	8.96	8.34	7.34	6.74
150	13.25	12.39	10.54	9.17
200	20.38	16.83	14.75	12.47
250	26.34	21.16	18.61	16.31
300	33.66	27.31	24.36	20.79

**Table 5 sensors-21-02049-t005:** Latency time for IoT information processing in a blockchain-based overlay network.

Number of IoT Info.	Latency Time for IoT Information Processing (ms)		
	Common Network	Overlay Network	Overlay Network Based on Blockchain
100	73.25	58.74	46.29
250	66.57	52.69	43.48
500	61.15	48.53	40.26
750	58.83	46.75	39.27
1000	57.18	45.03	37.74

**Table 6 sensors-21-02049-t006:** Comparison of integrity verification processing times by blockchain size.

Number of IoT Info.	Integrity Verification Processing Times by Blockchain Size (ms)		
	Common Network	Overlay Network	Overlay Network Based on Blockchain
100	13.74	11.21	9.91
250	15.67	13.31	10.37
500	18.31	15.76	13.32
750	21.08	18.65	17.84
1000	25.17	22.39	19.08

**Table 7 sensors-21-02049-t007:** Blockchain-based integrity verification overhead comparison.

Number of IoT Info.	Latency Time for IoT Information Processing (ms)					
	No Overlay Network			Overlay Network		
	No Using RSA	Only Using RSA	Using TCA and RSA	No Using RSA	Only Using RSA	Using TCA and RSA
100	63.19	57.81	54.82	60.79	54.72	51.71
250	67.71	63.94	57.65	64.67	57.69	53.27
500	71.54	67.62	60.74	66.54	60.92	57.62
750	76.13	72.64	65.93	71.32	64.81	54.58
1000	81.32	77.63	69.78	76.14	67.34	60.42

TCA: Top Certificate Authority; RCA: Region Certificate Authority.

**Table 8 sensors-21-02049-t008:** Comparison of blockchain-based integrity verification accuracy.

Number of IoT Info.	Verification Accuracy for IoT Information Processing (ms)					
	No Overlay Network			Overlay Network		
	No Using RSA	Only Using RSA	Using TCA and RSA	No Using RSA	Only Using RSA	Using TCA and RSA
100	63.94	67.93	73.98	70.19	73.17	76.93
250	66.76	69.86	75.19	72.91	75.31	78.84
500	70.03	72.64	77.25	73.94	77.83	81.16
750	73.85	75.04	80.37	76.75	80.16	84.75
1000	77.39	81.06	84.63	83.68	85.95	89.42

TCA: Top Certificate Authority; RCA: Region Certificate Authority.

**Table 9 sensors-21-02049-t009:** Rate of change according to blockchain length.

Evaluation Item	Key Length in Blockchain			
	64	128	256	512
$\bar{BK}$	53.148	53.967	54.525	54.742
ABKP(%)	41.521	42.161	42.597	42.767
∆BK	4.424	4.258	4.892	4.923
∆ABKP	4.017	3.879	4.325	4.415

$\bar{BK}$: Blockchain average key length; ABKP(%): Average blockchain key conversion probability; ∆BK: Converted blockchain key length standard variance; ∆ABKP: Converted average blockchain key distribution probability.

## Data Availability

Publicly available datasets were analyzed in this study. The data presented in this study are available on request from the corresponding author.

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
