# Peer review of "Hierarchical Multipath Blockchain Based IoT Information Management Techniques for Efficient Distributed Processing of Intelligent IoT Information"

_sensors, 2021, doi:10.3390/s21062049_

Round 1

Reviewer 1 Report

The authors of the paper know my previous thoughts about this article (#1 reviewer), so I would like to emphasize what happened since the last review process.

In my opinion, the paper has been evolved a lot. The extended Evaluation chapter is the most significant improvement, despite being the strongest part of the original draft too. It is clear, and the presented results and tests are exhausting; it does not only support the conclusions but also highlights the significance of the proposed concept. The improvements of chapter 3 are also fundamental; they make it easier to follow the main idea through the chapter - both the additional text and figures with higher resolution.

However, the authors added new paragraphs to chapter 2 (related works), which cite recent publications in the topic of data integrity, but for me, that is not enough to consider this chapter as “well-written”. I am sure that the authors of the paper know the recent scientific literature and state-of-the-art solutions deeply, but the purpose of the related works section is to concentrate the knowledge for the reader and, most importantly, the influence of these publications on this particular work. Nonetheless, the chapter's current state may be sufficient for experts in this field, but I think such an article should target a broader audience.

Author Response

Thank you for your opinion. I insert to section 2.2.4 added the following for access by the broader audience.

- Y. Dong et al. [29] conducted a protocol-related study for data integrity check in a cloud environment. Dong et al. analyzes the requirements and problems of state-of-the-art protocols for integrity checks of data in cloud environments. Furthermore, we compare the advantages and disadvantages of protocols used in cloud environments with each other.
S. R. Pujar et al. studied data integrity and validation, including data integrity checks, performance metrics, security attacks and updates for cloud storage [30]. Specifically, the taxonomy developed based on security-level provisioning used in various techniques for data integrity and validation techniques provides a global view of the problem and solution.
K. U. Maheswari et al. extends the cloud platform to study various integrity verification methods for IoT applications to be applied to both cloud and fog environments [31]. In particular, K. U. Maheswari et al. describes the authentication structure and its limitations being used in existing integrity verification protocols.

Reviewer 2 Report

Dear authors,

Congratulations on the improvements made in the manuscript. I believe that the manuscript has achieved enough quality to be published in this prestigious journal.

Author Response

Thank you for reviewer review

Reviewer 3 Report

Paper deals with important task. Authors peroposed a blockchain-based integrity verification technique in which large amounts of IoT information processed in distributed cloud environments can be guaranteed integrity in security threats related to IoT systems and CPS.

Paper has a great practical value.

Suggestions:

  1. Authors provide what they do (lines 60-62) and what it gives (line 64 ....). However, Authors should also write how it is done, either on the basis of what or using what
  2. All figures are very small. Please fix it
  3. About the references: line 210 - the reference is correct, line 2019 - the reference is incorrect. I cant fint the reference [28] in the main text of the paper. Please be careful and fix it

Author Response

Point 1: Authors provide what they do (lines 60-62) and what it gives (line 64 ....). However, Authors should also write how it is done, either on the basis of what or using what

Response 1: As reviewer request,

- I modified the sentence as below so that the meaning transmission is clear in line 60-62.

[before]

In this paper, we propose a blockchain-based IoT big data integrity verification technique that can ensure the integrity of large amounts of information sent and received by heterogeneous IoT devices in distributed cloud environments.

[after]

In this paper, we propose a blockchain-based IoT big data integrity verification technique that can ensure the integrity of IoT information so that large amounts of information sent and received from heterogeneous IoT devices are safely sent to cloud servers (data centers).

- The IoT device information described in line 64 refers to the urgent data that needs to be immediately worked on and the data sent to the data center for subsequent measurement and use.

Based on the above contents, I modified the sentence in line 64 as below.

[before]

The proposed technique ensures the integrity of IoT information by linking information from non-neighboring specific IoT devices to blockchain-based multiple hash chains.

[after]

The proposed technique ensures the integrity of IoT information by connecting information from specific non-adjacent IoT devices (emergency data that needs to be immediately undertaken and data sent to the data center for subsequent measurement and use) to blockchain-based multiple hash chains.

Point 2: All figures are very small. Please fix it

Response 2: I fixed all the pictures the same size.

Point 3: About the references: line 210 - the reference is correct, line 2019 - the reference is incorrect. I cant fint the reference [28] in the main text of the paper. Please be careful and fix it

Response 3: I marked the reference [26] incorrectly on line 217 and modified it as follows.

[before]

line 217 : Ref. [26]

[after]

Line 219 : Ref. [27]

[before]

line 223 : Ref. [27]

[after]

Line 225-> Ref. [28]
